# A Novel Bacterial 6-Phytase Improves Growth Performance, Tibia Mineralization and Precaecal Digestibility of Phosphorus in Broilers: Data from Four Independent Performance Trials

Maamer Jlali [1,*], Clémentine Hincelin [2], Marta I. Gracia [3], Farina Khattak [4], Maria Francesch [5], Tania Rougier [1], Pierre Cozannet [1], Guillermo Cano López [3], Marcio Ceccantini [2], Baris Yavuz [2], Sarper Ozbek [1], Aurélie Preynat [1] and Estelle Devillard [1]

[1] Adisseo France S.A.S., Center of Expertise in Research and Nutrition, 03600 Malicorne, France; tania.rougier@adisseo.com (T.R.); pierre.cozannet@adisseo.com (P.C.); sarper.ozbek@adisseo.com (S.O.); aurelie.preynat@adisseo.com (A.P.); estelle.devillard@adisseo.com (E.D.)
[2] Adisseo France S.A.S., 92160 Antony, France; clementine.hincelin@adisseo.com (C.H.); marcio.ceccantini@adisseo.com (M.C.); baris.yavuz@adisseo.com (B.Y.)
[3] Imasde Agroalimentaria, S.L. C/Nápoles 3, 28224 Pozuelo de Alarcón, Madrid, Spain; mgracia@e-imasde.com (M.I.G.); gcano@e-imasde.com (G.C.L.)
[4] Monogastric Science Research Centre, Scotland's Rural College, Easter Bush, Edinburgh EH25 9RG, Scotland, UK; farina.khattak@sruc.ac.uk
[5] Institute of Agrifood Research and Technology, Ctra. Reus-El Morell km. 3.8, 43120 Constantí, Tarragona, Spain; maria.francesch@irta.cat
* Correspondence: maamer.jlali@adisseo.com; Tel.: +33-6-40-68-57-92

**Abstract:** A series of four broiler performance studies were conducted in different facilities to investigate the efficacy of a novel bacterial 6-phytase added at 500 FTU/kg diet on growth performance, bone mineralization and precaecal digestibility of phosphorus (pcdP) in broilers fed diets deficient in available P (avP) and calcium (Ca). The experimental design was the same for all studies, with each having three treatments: positive control (PC) diet formulated to meet or exceed the requirements of birds, negative control (NC) diet similarly reduced by 0.15% points in avP and Ca compared to the PC diet, and the NC diet supplemented with phytase (PHY) at 500 FTU/kg diet from 1 to 35 days of age. Body weight (BW) and feed intake were measured at 21 and 35 days of age, and average daily gain (ADG), average daily feed intake (ADFI), feed conversion ratio (FCR), BW gain-corrected-FCR (cFCR), mortality and European performance efficiency factor (EPEF) were calculated. Tibia dry matter, tibia ash content and pcd of P were measured at 21 days of age in all experiments. The analysis of the data from the four experiments showed that compared with birds fed the adequate-nutrient diet, birds fed the NC diet resulted in a decrease ($p < 0.05$) in BW, ADG, ADFI and EPEF by 6.4, 6.3, 5.9 and 7.1%, respectively, and an increase in ($p = 0.02$) cFCR by 2.0%. The tibia dry matter and tibia ash content of these birds were also reduced ($p < 0.001$) by 3.8 and 4.0% points, respectively. PHY diets improved ($p < 0.05$) BW, ADG, ADFI, EPEF and cFCR by 8.0, 8.3, 7.3, 10.6 and 2.8%, respectively. Phytase addition at 500 FTU/kg diet also increased ($p < 0.001$) the tibia dry matter and tibia ash content by 3.5 and 4.2% points, respectively. The pcd of P was improved ($p < 0.001$) by 11.1 and 11.3% points, in comparison with NC and PC diets, respectively, when phytase was added. These performance parameters and tibia mineralization obtained with a diet supplemented with phytase were comparable to or better than the PC diet. The results demonstrated that avP and Ca could be lowered similarly by 0.15% points in broilers diets by using the new bacterial 6-phytase at 500 FTU/kg diet.

**Keywords:** broilers; digestibility; mineralization; performance; phytase

## 1. Introduction

Phosphorus (P) is involved in several chemical reactions in the body and acts directly on the growth, development and maintenance of skeleton and muscle. However, up to

80% of phosphorus (P) is present as phytate-P, bound to the phytic acid molecule in plant-based feed ingredients and unavailable for animals not equipped with dedicated enzymes. In addition, phytate can form complexes with protein, amino acids and some minerals, causing negative impacts on these nutrient's availability and consequently impairing growth performance, feed efficiency and bone mineralization [1–3]. Therefore, phytate-degrading enzymes have been included in broiler diets for over 30 years. Indeed, phytases are used to liberate phytate-bound P, thus improving the availability of other nutrients resulting in reduced feed costs and P excretion in the environment [3–5].

Indeed, the use of phytase allowed to reduce P and Ca levels in broiler diets supplemented with exogenous phytase [6]. The exogenous phytases commonly used in monogastric nutrition can be classified as 3-phytase and 6-phytase having different modes of action, substrate affinity and initiating degradation either by carbon 3 or 6, respectively [7–10]. The efficacy of exogenous phytase in monogastric animals is generally assessed through its positive effects on performance, bone mineralization and digestibility of P. It is well known that phytase addition in poultry diets improved not only the availability of P, but also of amino acids and Ca and therefore the growth performance [11–13]. As reported by Selle et al. [3], the major site of the activity of bacterial phytase in the gastrointestinal tract (GIT) of broilers is the gizzard, where digesta retention in this organ would facilitate phytate breakdown.

A new biosynthetic bacterial 6-phytase produced by *Trichoderma reesei* was recently developed. This novel 6-phytase is characterized with a pH profile ranging from 2.5 to 6.5, a high affinity for myo-inositol hexakisphosphate (IP6) and other myo-inositol phosphate (IP) esters, and a high intrinsic thermostability. However, limited information is available about the efficacy and consistency of this new phytase addition at standard dose (500 FTU/kg diet) in broilers. Therefore, the efficacy and consistency of a novel bacterial 6-phytase on growth performance, tibia ash and P contents and precaecal digestibility (pcd) of P in broilers from 1 to 35 days of age was evaluated over four trials reared in different experimental facilities and fed different feed ingredients.

## 2. Materials and Methods

Four growth experiments were conducted with broiler chicks in four different facilities. All animals were reared according to protocols approved by the Animal Care and Use committees at IRTA Animal Nutrition (Tarragona, Spain), IMASDE Agroalimentaria (Madrid, Spain), Center of Expertise in Research and Nutrition (Malicorne, France) and Scotland's Rural College (Edinburgh, Scotland, UK).

### 2.1. Enzyme Product

The phytase used in all experiments is a novel biosynthetic bacterial 6-phytase (Rovabio PhyPlus, Adisseo France S.A.S, Antony, France) derived from *Buttiauxella* sp. and produced by *Trichoderma reesei*. The pH profile of this new phytase ranges from 2.5 to 6.0, with a high affinity for IP6 and other IP esters and a high intrinsic thermostability. The phytase was applied in liquid form after pelleting to provide 500 phytase units (FTU)/kg of feed. One FTU is the amount of enzyme that liberates one micromole of inorganic orthophosphate from phytic acid per minute at pH 5.5 and 37 °C [14].

### 2.2. Experimental Design

The main specificities of the performance trials are summarized in Table 1. The experimental design was the same for all studies, with three treatments: a positive control (PC) diet formulated to meet or exceed the requirements of birds (starter phase from d1 to d21: ME = 3100 kcal/kg, dig. Lys = 1.22%, Ca = 0.90% and avP = 0.41% and grower phase from d22 to d35: ME = 3150 kcal/kg, dig. Lys = 1.01%, Ca = 0.60% and avP = 0.32%), a negative control (NC) diet reduced similarly by 0.15% points in avP and Ca compared to the PC diet, and the NC diet supplemented with phytase at 500 FTU/kg diet (PHY) from 0 to 35 days of age. The same feeding program with starter phase from d1 to d21 and

grower phase from d22 to d35 was used in the four studies. In addition, titanium dioxide was included at 5 g/kg in all diets as an indigestible marker to determine the pcd of P.

**Table 1.** Main specificities of performance trials.

| | Experiment 1 | Experiment 2 | Experiment 3 | Experiment 4 |
|---|---|---|---|---|
| Country | Spain (IRTA) | Spain (IMASDE) | France (CERN) | UK (SRUC) |
| Broiler strain | Ross 308 | Ross 308 | Ross 308 | Ross 308 |
| Gender | Male | Male | Male | Male |
| Basal diet | Corn-SBM | Corn-SBM | Corn-wheat-SBM | Corn-SBM |
| Treatments | | PC diet<br>NC diet = PC similarly reduced in avP and Ca by 0.15% unit<br>NC diet + Phytase at 500 FTU/kg diet | | |
| Inorganic P source | DCP | MCP | MCP | MCP |
| Nb rep/nb birds/trt | 16 rep × 45 birds | 24 rep × 18 birds | 16 rep × 25 birds | 24 rep × 18 birds |
| Duration | | 35 d | | |
| Measurements | | Body weight (BW), average daily gain (ADG), average daily feed intake (ADFI), feed conversion ratio (FCR) at 21 and 35 days of age.<br>BW gain-corrected-FCR (cFCR) and European production efficiency factor (EPEF) at 35 days of age.<br>Tibia dry matter, tibia ash content and precaecal digestibility of phosphorus at 21 days of age. | | |

PC: positive control diet, NC: negative control diet; DCP: dicalcium phosphate, MCP: monocalcium phosphate.

Experiment 1

A total of 2160 day-old Ross 308 male chicks were obtained from a local hatchery (Granja El Pilar, l'Aldea, Spain). They were allocated to 3 dietary treatments, replicated 16 times each with 45 chickens per replicate pen, in a randomized complete block design. Birds were randomly distributed into 48 pens with fresh wood shavings as bedding ($1.97 \times 1.98$ m$^2$ each). Each pen was equipped with two tub feeders and one circular drinker, with an available surface of 3.60 m$^2$. The temperature program was adjusted according to the standard program used in the farm: at 32–34 °C from 0 to 2 d, at 29–31 °C from 3 to 7 d, at 26–28 °C from 8 to 14 d, at 23–25 °C from 14 to 21 d, at 20–22 °C from 22 to 28 d and at 19–21 °C afterwards. The standard lighting program was 24 h of light for 2 days, 18 h of light and 6 h of dark per day for 7 days, and 16 h of light and 8 h of dark until the end of the experiment. All diets were mixed at IRTA (Constantí, Tarragona, Spain). Diets were based on corn and soybean meal. The composition and the estimated nutrient content of basal experimental diets are presented in Table 2. The diets were in the form of crumbs from 1 to 7 d and in pellets (3 mm) afterwards, and water was provided ad libitum throughout the experimental period. Diets contained 50 g narasin plus 50 g nicarbazin/ton of feed as coccidiostat.

**Table 2.** Composition and nutrient characteristics of experimental diets used in Experiment 1 and Experiment 2.

| | Experiment 1 | | | | Experiment 2 | | | |
|---|---|---|---|---|---|---|---|---|
| | 1 to 21 d | | 22 to 35 d | | 1 to 21 d | | 22 to 35 d | |
| | PC | NC | PC | NC | PC | NC | PC | NC |
| Composition, % | | | | | | | | |
| Corn | 52.91 | 54.39 | 65.06 | 66.40 | 44.70 | 46.22 | 57.40 | 58.91 |
| Extruded soybean | 10.00 | 7.63 | - | - | 4.00 | 4.00 | - | - |
| Soybean meal 48% | 29.50 | 31.08 | 28.17 | 27.93 | 40.73 | 40.48 | 34.30 | 34.05 |
| Soy oil | 3.31 | 3.30 | 4.06 | 3.64 | 6.40 | 5.90 | 5.66 | 5.17 |
| Dicalcium phosphate | 1.60 | 0.75 | 1.19 | 0.33 | - | - | - | - |
| Monocalcium phosphate | - | - | - | - | 1.37 | 0.64 | 1.01 | 0.29 |
| Calcium carbonate | 0.54 | 0.73 | 0.11 | 0.29 | 1.05 | 1.00 | 0.52 | 0.46 |
| Salt | 0.36 | 0.37 | 0.36 | 0.36 | 0.23 | 0.23 | 0.24 | 0.24 |

**Table 2.** *Cont.*

| | Experiment 1 | | | | Experiment 2 | | | |
|---|---|---|---|---|---|---|---|---|
| | 1 to 21 d | | 22 to 35 d | | 1 to 21 d | | 22 to 35 d | |
| | PC | NC | PC | NC | PC | NC | PC | NC |
| Choline chloride | 0.06 | 0.06 | 0.05 | 0.05 | - | - | - | - |
| Sodium bicarbonate | - | - | - | - | 0.12 | 0.12 | 0.12 | 0.12 |
| DL-methionine 99% | 0.38 | 0.38 | 0.24 | 0.24 | 0.37 | 0.37 | 0.24 | 0.24 |
| L-lysine HCl 98% | 0.20 | 0.20 | 0.19 | 0.19 | 0.09 | 0.09 | 0.09 | 0.10 |
| L-threonine 98.5% | 0.07 | 0.06 | 0.05 | 0.05 | 0.03 | 0.03 | 0.02 | 0.02 |
| L-tryptophan | 0.03 | 0.03 | 0.02 | 0.02 | - | - | - | - |
| L-valine 96.5% | 0.06 | 0.06 | 0.02 | 0.02 | 0.02 | 0.02 | - | - |
| Premix [1,2] | 0.48 | 0.48 | 0.48 | 0.48 | 0.40 | 0.40 | 0.40 | 0.40 |
| Titanium dioxide | 0.50 | 0.50 | - | - | 0.50 | 0.50 | - | - |
| Calculated nutrients, % | | | | | | | | |
| ME, kcal/kg | 3100 | 3100 | 3150 | 3150 | 3100 | 3100 | 3150 | 3150 |
| Crude protein | 21.90 | 21.90 | 18.50 | 18.50 | 23.60 | 23.60 | 20.00 | 20.00 |
| Crude fat | 7.48 | 7.10 | 6.72 | 6.34 | 8.88 | 9.33 | 7.67 | 8.12 |
| Crude fiber | 2.71 | 2.72 | 2.49 | 2.51 | 3.36 | 3.34 | 3.03 | 3.01 |
| Ash | 6.18 | 5.51 | 4.86 | 4.19 | 5.07 | 5.69 | 3.81 | 4.43 |
| Dig. lysine | 1.22 | 1.22 | 1.01 | 1.01 | 1.22 | 1.22 | 1.01 | 1.01 |
| Dig. methionine | 0.68 | 0.67 | 0.50 | 0.49 | 0.68 | 0.68 | 0.51 | 0.51 |
| Dig. Met + Cys | 0.98 | 0.98 | 0.77 | 0.77 | 0.98 | 0.98 | 0.77 | 0.77 |
| Calcium | 0.90 | 0.75 | 0.60 | 0.45 | 0.90 | 0.75 | 0.60 | 0.45 |
| Total P | 0.66 | 0.51 | 0.55 | 0.40 | 0.71 | 0.55 | 0.60 | 0.44 |
| Available P | 0.41 | 0.26 | 0.32 | 0.17 | 0.41 | 0.26 | 0.32 | 0.17 |
| Sodium | 0.14 | 0.14 | 0.14 | 0.14 | 0.14 | 0.14 | 0.14 | 0.14 |

PC: positive control diet; NC = PC diet similarly reduced in available phosphorus and calcium by 0.15% unit. [1] The premix used in Experiment 1 provided per kg feed: vitamin A, 10,000 UI; vitamin D3, 4800 UI; vitamin E, 45 mg; vitamin B1, 3 mg; vitamin B2, 9 mg; vitamin B6, 4.5 mg; vitamin B12, 40 μg; vitamin K3, 3 mg; calcium pantothenate, 16.5 mg; niacin, 51 mg; folic acid, 1.8 mg; biotin, 0.15 mg; Fe, 54 mg; I, 1.2 mg; Cu, 12 g; Mn, 90 mg; Zn, 66 mg; Se, 0.18 mg. [2] The premix used in Experiment 2 provided per kilogram of diet: vitamin A, 9000 IU; vitamin D3, 2000 IU; vitamin E, 30.0 mg; vitamin B12, 12.0 μg; vitamin B6, 2.0 mg; vitamin K3, 2.0 mg; vitamin B1, 1.0 mg; vitamin B2, 5.0 mg; nicotinic acid, 30.0 mg; pantothenic acid, 10.0 mg; folic acid, 1.0 mg; biotin, 50.0 mg; choline chloride, 400 mg; Fe, 20.0 mg; Mn, 100.0 mg; Se, 0.2 mg; Zn, 80.0 mg; Cu, 10.0 mg; I, 2.0 mg.

Experiment 2

A total of 1296 day-old male Ross 308 chicks were obtained from a local hatchery. They were allocated to 3 dietary treatments, replicated 24 times each with 18 chickens per replicate pen, in a randomized complete block design. Chickens were randomly distributed into 72 pens ($1.30 \times 1.50$ m$^2$ each) with fresh wood shavings as bedding. The experimental facility was supplied with an automated biomass heating system, natural ventilation and artificial and programmable lights. The temperature was set at 32 °C at the trial start and decreased by 3 °C each week. Then, it was set at 23 °C from day 28 until the end of the experimental period. The lighting program was typically given 23 h of light at day old and then adjusted to 18 h light and 6 h dark. The experimental diets were formulated by IMASDE Agroalimentaria. The composition and the calculated analyses of the experimental diets are presented in Table 2. The tested phytase was premixed with ground cereal before adding it to the final mix to ensure homogeneity. Diets were provided in mash form, and water was ad libitum available throughout the experimental period.

Experiment 3

A total of 1200 day-old male Ross 308 chicks were obtained from a local hatchery (Auvergne Poussins, Allier, France). They were individually weighed and randomly allocated to 3 dietary treatments, replicated 16 times each with 25 chickens per replicate pen, in a randomized complete block design. Chickens were randomly distributed into 48 pens ($1.5 \times 1.0$ m$^2$ each). Each pen contained wood shavings as litter, one hanging feeder and one bell-shaped water trough. Room temperature for the chicks was initially set at

32 °C for the first week and gradually decreased to 23.4 °C at d35. From d1 to 7, birds were exposed to 23 h light, and after d7, the hours of light were decreased gradually until 4 h of darkness was reached on d14 until d35. All diets were manufactured at Euronutrition (Saint-Symphorien, France). Diets were based on corn, wheat and soybean meal. The composition and the estimated nutrient content of basal experimental diets are presented in Table 3. From 0 to 8 d, diets were provided as crumbs and pellets until the end of the experimental period. Diets and water were provided as ad libitum throughout the experimental period.

**Table 3.** Composition and nutrient characteristics of experimental diets used in Experiment 3 and Experiment 4.

|  | Experiment 3 | | | | Experiment 4 | | | |
|---|---|---|---|---|---|---|---|---|
|  | 1 to 21 d | | 22 to 35 d | | 1 to 21 d | | 22 to 35 d | |
|  | **PC** | **NC** | **PC** | **NC** | **PC** | **NC** | **PC** | **NC** |
| Composition, % | | | | | | | | |
| Corn | 20.00 | 20.00 | 20.00 | 20.00 | 51.03 | 52.43 | 65.46 | 66.84 |
| Wheat | 30.84 | 32.39 | 45.81 | 47.20 | - | - | - | - |
| Soybean meal 48% | 37.00 | 36.73 | 24.68 | 24.53 | 39.51 | 39.31 | 27.72 | 27.52 |
| Soy oil | 7.62 | 7.21 | 6.47 | 6.08 | 4.94 | 4.51 | 3.68 | 3.25 |
| Monocalcium phosphate | 1.33 | 0.53 | 0.88 | 0.08 | 1.24 | 0.57 | 0.93 | 0.27 |
| Calcium carbonate | 1.15 | 1.10 | 0.65 | 0.61 | 1.38 | 1.28 | 0.83 | 0.73 |
| DL-methionine 99% | 0.38 | 0.37 | 0.25 | 0.24 | 0.37 | 0.37 | 0.25 | 0.25 |
| L-lysine HCl 98% | 0.17 | 0.17 | 0.25 | 0.25 | 0.08 | 0.09 | 0.17 | 0.18 |
| L-threonine 98.5% | 0.08 | 0.08 | 0.10 | 0.10 | 0.07 | 0.07 | 0.08 | 0.08 |
| L-valine 96.5% | 0.09 | 0.08 | 0.07 | 0.07 | 0.07 | 0.07 | 0.07 | 0.07 |
| Salt | 0.34 | 0.34 | 0.34 | 0.34 | 0.35 | 0.35 | 0.35 | 0.35 |
| Premix [1,2] | 0.50 | 0.50 | 0.50 | 0.50 | 0.46 | 0.46 | 0.46 | 0.46 |
| Titanium dioxide | 0.50 | 0.50 | - | - | 0.50 | 0.50 | - | - |
| Calculated nutrients, % | | | | | | | | |
| ME, kcal/kg | 3100 | 3100 | 3150 | 3150 | 3100 | 3100 | 3150 | 3150 |
| Crude protein, % | 22.93 | 22.87 | 18.95 | 18.86 | 22.86 | 22.87 | 18.41 | 18.42 |
| Crude fat, % | 8.78 | 9.16 | 7.73 | 8.10 | 7.35 | 6.97 | 6.41 | 6.03 |
| Crude fiber, % | 2.63 | 2.61 | 2.46 | 2.44 | 3.51 | 3.53 | 3.12 | 3.14 |
| Ash, % | 5.25 | 5.93 | 3.87 | 4.54 | 5.68 | 5.05 | 4.35 | 3.72 |
| Dig. lysine, % | 1.22 | 1.22 | 1.01 | 1.01 | 1.22 | 1.22 | 1.01 | 1.01 |
| Dig. methionine, % | 0.67 | 0.67 | 0.50 | 0.50 | 0.68 | 0.68 | 0.51 | 0.51 |
| Dig. Met + Cys, % | 0.98 | 0.98 | 0.77 | 0.77 | 0.98 | 0.98 | 0.77 | 0.77 |
| Calcium, % | 0.90 | 0.75 | 0.60 | 0.45 | 0.90 | 0.75 | 0.60 | 0.45 |
| Total P, % | 0.70 | 0.52 | 0.56 | 0.39 | 0.69 | 0.55 | 0.58 | 0.43 |
| Available P, % | 0.41 | 0.26 | 0.32 | 0.17 | 0.41 | 0.26 | 0.32 | 0.17 |
| Sodium, % | 0.14 | 0.14 | 0.14 | 0.14 | 0.14 | 0.14 | 0.14 | 0.14 |

PC: positive control diet; NC = PC diet similarly reduced in available P and Ca by 0.15% unit. [1] The premix used in Experiment 3 provided per kilogram of diet: vitamin A, 10,000 IU; vitamin D3, 4500 IU; vitamin E, 100 IU; vitamin B12, 20.0 μg; vitamin B6, 4.0 mg; vitamin K3, 3.0 mg; vitamin B1, 2.5 mg; vitamin B2, 8.0 mg; nicotinic acid, 40.0 mg; pantothenic acid, 20.0 mg; folic acid, 1.5 mg; biotin, 0.3 mg; choline chloride, 550 mg; Fe, 60.0 mg; Mn, 80.0 mg; Se, 0.2 mg; Zn, 80.0 mg; Cu, 15.0 mg; I, 1.0 mg. [2] The premix used in Experiment 4 provided per kilogram of diet: vitamin A, 10,000 IU; vitamin D3, 5000 IU; vitamin E: 100.0 mg; vitamin B12, 30.0 μg; vitamin B6, 3.0 mg; vitamin B1, 3.0 mg; vitamin B2, 10.0 mg; nicotinic acid, 60.0 mg; pantothenic acid, 15.0 mg; folic acid, 1.5 mg; biotin, 25.0 mg; choline chloride, 250 mg; Fe, 20.0 mg; Mn, 100.0 mg; Se, 0.25 mg; Zn, 80.0 mg; Cu, 10.0 mg; I, 1.0 mg.

Experiment 4

A total of 1296 day-old male Ross 308 chicks were obtained from a local hatchery. They were randomly allocated to 3 dietary treatments in a randomized complete block design. Each treatment consisted of 24 replicate pens with 18 birds per pen. Each pen (2.2 m$^2$) contained wood shavings as litter, with manual feeders (one feeder/pen), and an automatic drinking nipple line system was used. The experimental conditions were automatically controlled and appropriate for the age of the broilers. Temperature for the

chicks was initially set at 32 °C for the first week and gradually decreased to 21 °C at d35. Birds were typically given 23 h of light at day old and stepped down to 18 h light and 6 h dark by 4 days. All birds were fed crumbs during the starter phase (1 to 21 days) and pellets during the grower phase (22 to 35 days). Diets were based on corn and soybean meal. The composition and nutrient characteristics of experimental diets are presented in Table 3. Diets and water were provided as ad libitum throughout the experimental period.

### 2.3. Performance Measurement

Body weight (BW) and feed intake (FI) were measured at 1, 21 and 35 days of age, and average daily gain (ADG), average daily feed intake (ADFI), feed conversion ratio (FCR) and mortality were calculated at the end of each feeding phase and for the global experimental period. BW gain-corrected-FCR (cFCR) was calculated for the global experimental period of each experiment. On d 21, 5 birds per pen in experiment 1 (n = 240 birds) and 3 birds per pen in experiments 2 (n = 216 birds), 3 (n = 144 birds) and 4 (n = 216 birds) were randomly selected then sacrificed for ileal digesta collection. Birds were either euthanized by intravenous injection (0.2 mL per kg body weight) of sodium pentobarbital or culled via cervical dislocation. The digesta was gently flushed out from the lower half toward the ileocecal junction, pooled by pen and stored at −18 °C. In addition, the left tibias were collected (randomly chosen) from 1 bird per replicate in experiments 1 (n = 16 birds per treatment), 3 (n = 16 birds per treatment) and 4 (n = 16 birds per treatment) or from 2 birds per replicate (n = 48 birds per treatment) in experiment 2. Then, tibias collected were pooled by pen and stored at −20 °C until further analysis.

### 2.4. Chemical Analyses

Representative samples of all experimental diets were analyzed for dry matter, crude protein, ether extract, ash, Ca and total P, phytate-P and titanium dioxide. In addition, lyophilized ileal digesta samples for all experiments were analyzed for total P and titanium dioxide. Dry matter, crude fat, ash and nitrogen (N) were determined using the AOAC procedures [15,16]. Total P and Ca were analyzed by atomic absorption spectroscopy and UV-vis spectroscopy methods [12,13]. Phytate-P was analyzed by colorimetric (Kyoto, Japan) method according to Haug and Lantzch [17]. The concentration of titanium dioxide in diets and lyophilized ileal digesta samples were analyzed following the procedure of Short et al. [18] and Myers et al. [19]. The activity of phytase in the experimental diets were analyzed according to the ISO standard methodology [14]. Ash content in tibia bones was determined by removing the adhering tissue, drying the bone at 103–110 °C and extracting the fat with ether. The dry fat-free bones were ashed in a muffle furnace at 550 °C, according to AOAC [15], to measure ash content.

### 2.5. Calculation and Statistical Analysis

The pcd of P was calculated using titanium dioxide as the inert marker:

$$\text{pcd of P (\%)} = [1 - (\text{Ti}_{\text{diet}}/\text{Ti}_{\text{ileal digesta}}) \times (\text{P}_{\text{ileal digesta}}/\text{P}_{\text{diet}})] \times 100$$

where $\text{Ti}_{\text{diet}}$ is the titanium dioxide concentration in the diet, $\text{Ti}_{\text{ileal digesta}}$ is the titanium dioxide concentration in the ileal digesta, $\text{P}_{\text{ileal digesta}}$ is the concentration of P in the ileal digesta and $\text{P}_{\text{diet}}$ is the concentration of P in the diet.

European performance efficiency factor (EPEF) is a value that standardizes technical results, considering FCR, mortality/culling and BW, and it was calculated as follows:

$$\text{EPEF} = [(\text{average daily BWG (g)}) \times (\% \text{ survival rate})]/(\text{FCR} \times 10)$$

The data from each experiment were analyzed using GLM procedures of SAS software (SAS version 9.1.3 Cray, SAS Institute, Inc., Cary, NC, USA). Treatment and block were considered as the fixed effects. The following experimental model was used:

$$Yij = \mu + \alpha i + \beta j + \varepsilon ij,$$

where $Yij$ = the dependent variables; $\mu$ = general mean; $\alpha i$ = effect of dietary treatment; $\beta j$ = effect of block; and $\varepsilon ij$ = random error.

All data obtained from all experiments were analyzed using a mixed model. Treatment was considered as the fixed effect, and the study number was considered as the random effect. The pen was considered as the experimental unit. Least square means were compared using the Fisher test, and differences were considered significant when $p \leq 0.05$, and trends were noted when $0.05 < p < 0.10$.

## 3. Results

### 3.1. Experimental Diets

The analyzed values of dietary nutrients and phytase activities in experimental diets are presented in Tables 4 and 5, respectively. The analyzed crude protein, ether extract, ash, Ca, total P and phytate-*p* values were in agreement with the calculated values in Tables 2 and 3. Analyzed phytase activities in the experimental diets supplemented with phytase were within the range of the target value (500 FTU/kg diet) $\pm$ 20%.

**Table 4.** Analyzed nutrients in experimental diets of the starter phase (1 to 21 days of age).

| | Dry Matter, % | Crude Protein, % | Ether Extract, % | Ash, % | Total P, % | Phytic P, % | Ca, % | Phytase Activity, FTU/kg Diet |
|---|---|---|---|---|---|---|---|---|
| Experiment 1 (IRTA) | | | | | | | | |
| PC | 88.7 | 21.9 | 7.1 | 5.6 | 0.63 | 0.21 | 0.74 | 0 |
| NC | 88.8 | 21.7 | 7.0 | 5.3 | 0.48 | 0.22 | 0.69 | 71 |
| PHY | 88.8 | 21.5 | 7.0 | 5.4 | 0.50 | 0.20 | 0.67 | 506 |
| Experiment 2 (IMASDE) | | | | | | | | |
| PC | 89.1 | 24.0 | 8.8 | 6.3 | 0.68 | 0.24 | 0.95 | 64 |
| NC | 88.5 | 24.3 | 8.4 | 5.6 | 0.51 | 0.25 | 0.79 | 64 |
| PHY | 88.5 | 24.2 | 7.7 | 5.7 | 0.49 | 0.26 | 0.80 | 509 |
| Experiment 3 (CERN) | | | | | | | | |
| PC | 89.6 | 22.7 | 8.5 | 5.9 | 0.68 | 0.28 | 0.89 | 33 |
| NC | 89.5 | 22.7 | 8.1 | 5.4 | 0.48 | 0.30 | 0.73 | 83 |
| PHY | 89.6 | 23.0 | 9.1 | 5.5 | 0.50 | 0.29 | 0.72 | 567 |
| Experiment 4 (SRUC) | | | | | | | | |
| PC | 88.6 | 22.3 | 6.9 | 5.5 | 0.52 | 0.26 | 0.83 | 68 |
| NC | 88.9 | 23.4 | 6.9 | 5.7 | 0.50 | 0.28 | 0.76 | 108 |
| PHY | 89.2 | 21.5 | 7.3 | 5.8 | 0.49 | 0.26 | 0.84 | 521 |

PC: positive control diet; NC: negative control diet; PHY: NC diet supplemented with phytase (Rovabio PhyPlus) at 500 FTU/kg diet.

**Table 5.** Analyzed nutrients in experimental diets of grower phase (22 to 35 days of age).

| | Dry Matter, % | Crude Protein, % | Ether Extract, % | Ash, % | Total P, % | Phytic P, % | Ca, % | Phytase Activity, FTU/kg Diet |
|---|---|---|---|---|---|---|---|---|
| Experiment 1 (IRTA) | | | | | | | | |
| PC | 88.7 | 18.5 | 6.6 | 4.6 | 0.52 | 0.23 | 0.51 | 66 |
| NC | 88.7 | 18.6 | 6.2 | 4.2 | 0.38 | 0.23 | 0.41 | 51 |
| PHY | 88.6 | 18.6 | 6.1 | 4.2 | 0.38 | 0.22 | 0.40 | 474 |
| Experiment 2 (IMASDE) | | | | | | | | |
| PC | 88.5 | 20.8 | 7.5 | 5.1 | 0.59 | 0.25 | 0.86 | 63 |
| NC | 88.4 | 21.0 | 7.1 | 4.6 | 0.38 | 0.25 | 0.51 | 67 |
| PHY | 88.4 | 20.9 | 7.9 | 4.6 | 0.38 | 0.25 | 0.54 | 497 |

**Table 5.** *Cont.*

| | Dry Matter, % | Crude Protein, % | Ether Extract, % | Ash, % | Total P, % | Phytic P, % | Ca, % | Phytase Activity, FTU/kg Diet |
|---|---|---|---|---|---|---|---|---|
| Experiment 3 (CERN) | | | | | | | | |
| PC | 88.9 | 19.2 | 8.5 | 5.0 | 0.54 | 0.26 | 0.60 | 0 |
| NC | 89.1 | 18.7 | 7.8 | 4.3 | 0.36 | 0.27 | 0.48 | 188 |
| PHY | 89.3 | 19.5 | 7.4 | 4.3 | 0.36 | 0.27 | 0.42 | 527 |
| Experiment 4 (SRUC) | | | | | | | | |
| PC | 88.2 | 17.6 | 6.1 | 5.1 | 0.50 | 0.25 | 0.64 | 42 |
| NC | 88.5 | 17.9 | 6.3 | 4.9 | 0.40 | 0.24 | 0.59 | 57 |
| PHY | 88.7 | 16.9 | 6.3 | 4.8 | 0.38 | 0.24 | 0.57 | 402 |

PC: positive control diet; NC: negative control diet; PHY: NC diet supplemented with phytase (Rovabio PhyPlus) at 500 FTU/kg diet.

### 3.2. Individual Experiments

#### Experiment 1

The results of growth performance, tibia mineralization and pcd of P are presented in Table 6. Compared with birds fed the adequate-nutrient diet (PC), broilers fed the diet reduced in avP, and Ca had lower ($p < 0.0001$) final BW ($-7.2\%$), ADG ($-7.3\%$), ADFI ($-7.1\%$) and EPEF ($-10.6\%$). Mineral depletion also decreased the tibia dry-matter and ash content by 5.0 and 3.6% points, respectively. No significant difference was observed between PC and NC diets on pcd of P. Broilers fed PHY diet had higher final BW ($+11.4\%$, $p < 0.001$), ADG ($+11.6\%$, $p < 0.001$), ADFI ($+11.9\%$, $p < 0.001$) and EPEF ($+14.5\%$, $p < 0.001$) compared to those in NC group. In addition, birds fed the diet supplemented with phytase exhibited higher ($p < 0.001$) tibia dry matter by 4.7 and 5.7% points, respectively, than birds fed NC diet. The addition of phytase also improved ($p < 0.001$) the pcd of P by 18.6% points compared to the NC diet.

**Table 6.** Effect of phytase supplementation on growth performance, tibia traits and precaecal digestibility of phosphorus (Experiment 1 and Experiment 2).

| | Experiment 1 | | | | | Experiment 2 | | | | |
|---|---|---|---|---|---|---|---|---|---|---|
| | PC | NC | PHY [1] | SEM | *p*-Value | PC | NC | PHY [1] | SEM | *p*-Value |
| Growth performance | | | | | | | | | | |
| BW d1, g | 44 | 45 | 45 | - | 0.49 | 45.5 | 45.5 | 45.5 | - | 1.00 |
| BW d21, g | 1012 [b] | 924 [c] | 1064 [a] | 5.75 | <0.001 | 878 [a] | 834 [b] | 874 [a] | 11.18 | 0.01 |
| ADG 1–21 d, g/bird/d | 46.08 [b] | 41.85 [c] | 48.53 [a] | 0.27 | <0.001 | 39.47 [a] | 37.31 [b] | 39.21 [a] | 0.55 | 0.01 |
| ADFI 1–21 d, g/bird/d | 55.40 [b] | 50.34 [c] | 57.56 [a] | 0.29 | <0.001 | 49.90 [a] | 47.60 [b] | 50.70 [a] | 0.53 | 0.001 |
| FCR 1–21 d | 1.203 [a] | 1.203 [a] | 1.186 [b] | 0.004 | 0.005 | 1.266 | 1.278 | 1.297 | 0.014 | 0.30 |
| Mortality 1–21 d, % | 0.42 | 1.68 | 1.11 | 0.42 | 0.13 | 2.08 | 1.85 | 1.85 | 0.67 | 0.96 |
| ADG 22–35 d, g/bird/d | 109.86 [a] | 103.08 [b] | 112.35 [a] | 0.94 | <0.001 | 95.29 [a] | 87.61 [b] | 95.93 [a] | 1.09 | <0.001 |
| ADFI 22–35 d, g/bird/d | 162.45 [b] | 152.80 [c] | 169.14 [a] | 1.12 | <0.001 | 147.26 [a] | 135.76 [b] | 148.25 [a] | 1.41 | <0.001 |
| FCR 22–35 d | 1.479 [b] | 1.483 [b] | 1.506 [a] | 0.23 | 0.007 | 1.547 | 1.554 | 1.546 | 0.01 | 0.84 |
| Mortality 22–35 d, % | 1.834 | 3.88 | 2.09 | 0.62 | 0.06 | 1.39 | 1.73 | 0.56 | 0.68 | 0.46 |
| BW d35, g | 2550 [b] | 2367 [c] | 2637 [a] | 15.49 | <0.001 | 2219 [a] | 2066 [b] | 2218 [a] | 22.95 | <0.001 |
| ADG 1–35 d, g/bird/d | 71.59 [b] | 66.34 [c] | 74.06 [a] | 0.44 | <0.001 | 59.20 [a] | 55.02 [b] | 59.31 [a] | 0.69 | <0.001 |
| ADFI 1–35 d, g/bird/d | 97.60 [b] | 90.64 [c] | 101.44 [a] | 0.53 | <0.001 | 84.29 [a] | 78.63 [b] | 85.28 [a] | 0.76 | <0.001 |
| FCR 1–35 d | 1.363 | 1.366 | 1.37 | 0.003 | 0.32 | 1.425 | 1.432 | 1.439 | 0.009 | 0.53 |
| cFCR 1–35 d | 1.360 [b] | 1.381 [a] | 1.359 [b] | 0.004 | 0.001 | 1.412 [b] | 1.458 [a] | 1.426 [ab] | 0.014 | 0.05 |
| Mortality 1–35 d, % | 2.22 [b] | 5.47 [a] | 3.19 [b] | 0.78 | 0.02 | 3.24 | 3.24 | 2.31 | 0.85 | 0.67 |
| EPEF | 511 [a] | 457 [b] | 523 [a] | 5.42 | <0.001 | 400 [a] | 370 [b] | 403 [a] | 9.22 | 0.03 |
| Tibia traits at 21 d of age | | | | | | | | | | |
| Tibia DM, % | 44.00 [a] | 39.01 [b] | 43.70 [a] | 0.41 | <0.001 | 35.76 [a] | 33.81 [b] | 35.93 [a] | 0.24 | <0.001 |
| Tibia ash content, % DM | 48.43 [b] | 44.83 [c] | 50.52 [a] | 0.36 | <0.001 | 33.92 [a] | 30.35 [b] | 34.43 [a] | 0.41 | <0.001 |
| pcd of P at 21 d of age, % | 64.18 [b] | 62.08 [b] | 80.71 [a] | 0.86 | <0.001 | 60.45 [b] | 59.44 [b] | 65.77 [a] | 1.00 | <0.001 |

[a,b,c] Within a row, means without a common superscript letter differ ($p < 0.05$). PC: positive control diet; NC: negative control; PHY: negative control + 500 FTU phytase/kg diet; SEM: Standard error mean; BW: body weight, ADG: average daily gain; ADFI: average daily feed intake; FCR: feed conversion ratio; cFCR: BW gain-corrected-feed conversion ratio; EPEF: European Production Efficiency Factor; DM: dry matter. pcd of P: precaecal digestibility of phosphorus; [1] Phytase was supplemented at 500 FTU/kg diet.

Experiment 2

The results of all performance parameters are presented in Table 6. Broilers fed the NC diet having reduced avP and Ca had lower final BW (−6.9%, $p < 0.001$), ADG (−7.1%, $p < 0.001$), ADFI (−6.7%, $p < 0.001$) and EPEF (−7.4%, $p = 0.03$) than birds fed the PC adequate-nutrient diet. Additionally, birds fed the NC diet had decreased the tibia dry matter (−2.0% points, $p < 0.001$) and tibia ash content (−3.6% points) and did not influence ($p = 0.76$) the pcd of P in comparison to birds fed the PC diet. The phytase supplementation improved ($p < 0.001$) final BW, ADG, ADFI and EPEF by 7.4, 7.8, 8.5 and 9.0%, respectively, compared with birds fed the NC diet. In addition, birds fed the phytase-supplemented diet exhibited higher ($p < 0.001$) tibia dry matter and tibia ash content than birds fed the PC diet. The pcd of P was higher ($p < 0.01$) in broilers fed PHY diet by 6.6 and 6.8% points than in birds fed the PC and NC diets, respectively.

Experiment 3

The effects on growth performance, bone mineralization and pcd of P are shown in Table 7. Available P and Ca reduction decreased ($p < 0.001$) final BW, ADG, ADFI and EPEF by 10.6, 12.0, 10.7 and 12.8%, respectively, compared with PC-fed birds. In addition, tibia dry matter (−7.2% points, $p < 0.001$) and tibia ash content (−6.4% points, $p < 0.001$) were lower in NC fed than in birds. No significant difference was observed in pcd of P between PC and NC fed birds. Phytase supplementation improved the final BW, ADG, ADFI and EPEF by 10.6, 10.8, 12.3 and 12.6%, respectively. In addition, tibia dry matter, tibia ash content and pcd of P were improved ($p < 0.001$) by 5.6, 5.0 and 13.1% points, respectively, in broilers fed the diet supplemented with phytase than NC fed birds.

**Table 7.** Effect of phytase supplementation on growth performance, tibia traits and precaecal digestibility of phosphorus (Experiment 3 and Experiment 4).

| | Experiment 3 | | | | | Experiment 4 | | | | |
|---|---|---|---|---|---|---|---|---|---|---|
| | PC | NC | PHY [1] | SEM | *p*-Value | PC | NC | PHY [1] | SEM | *p*-Value |
| Growth performance | | | | | | | | | | |
| BW d1, g | 43 | 43 | 43 | - | 1.00 | 37 | 37 | 37 | - | 0.44 |
| BW d21, g | 1076 [b] | 958 [c] | 1094 [a] | 6.13 | <0.001 | 972 [b] | 950 [c] | 994 [a] | 5.12 | <0.001 |
| ADG 1–21d, g/bird/d | 49.19 [b] | 43.62 [c] | 50.08 [a] | 0.29 | <0.001 | 44.15 [b] | 43.33 [b] | 45.37 [a] | 4.9 | <0.001 |
| ADFI 1–21d, g/bird/d | 58.98 [a] | 52.84 [b] | 59.91 [a] | 0.43 | <0.001 | 58.22 | 59.82 | 57.67 | 0.87 | 0.21 |
| FCR 1–21d | 1.199 | 1.212 | 1.196 | 0.006 | 0.22 | 1.320 [ab] | 1.382 [a] | 1.271 [b] | 0.02 | 0.001 |
| Mortality 1–21d, % | 1.00 | 0.75 | 1.01 | 0.5 | 0.92 | 2.55 | 1.16 | 1.62 | 0.53 | 0.18 |
| ADG 22–35d, g/bird/d | 110.04 [a] | 98.65 [c] | 106.97 [b] | 1.22 | <0.001 | 108.74 [b] | 106.55 [b] | 111.58 [a] | 0.74 | <0.001 |
| ADFI 22–35d, g/bird/d | 161.44 [a] | 143.88 [b] | 160.71 [a] | 1.07 | <0.001 | 168.26 [a] | 161.35 [b] | 167.02 [ab] | 1.69 | 0.01 |
| FCR 22–35d | 1.467 [b] | 1.460 [b] | 1.509 [a] | 0.011 | 0.01 | 1.548 | 1.515 | 1.498 | 0.01 | 0.06 |
| Mortality 22–35d, % | 1.14 [b] | 4.02 [a] | 0.45 [b] | 0.7 | 0.003 | 1.65 | 0.48 | 0.71 | 0.42 | 0.13 |
| BW d35, g | 2616 [a] | 2340 [b] | 2588 [a] | 17.48 | <0.001 | 2506 [b] | 2445 [c] | 2559 [a] | 13.62 | <0.001 |
| ADG 1–35d, g/bird/d | 73.53 [a] | 65.63 [b] | 72.72 [a] | 0.5 | <0.001 | 66.28 [b] | 65.23 [b] | 68.28 [a] | 0.58 | 0.002 |
| ADFI 1–35d, g/bird/d | 99.96 [a] | 89.26 [b] | 100.23 [a] | 0.6 | <0.001 | 95.85 | 95.02 | 95.48 | 1.08 | 0.86 |
| FCR 1–35d | 1.359 | 1.36 | 1.379 | 0.007 | 0.13 | 1.446 [a] | 1.458 [a] | 1.399 [b] | 0.01 | 0.005 |
| cFCR 1–35d | 1.357 | 1.362 | 1.377 | 0.007 | 0.17 | 1.445 [a] | 1.447 [a] | 1.380 [b] | 0.01 | <0.001 |
| Mortality 1–35d, % | 2.00 | 4.25 | 1.39 | 0.85 | 0.06 | 4.17 [a] | 1.62 [b] | 2.31 [ab] | 0.7 | 0.036 |
| EPEF | 530 [a] | 462 [b] | 520 [a] | 4.62 | <0.001 | 440 [b] | 442 [b] | 477 [a] | 7.03 | 0.001 |
| Tibia traits at 21 d of age | | | | | | | | | | |
| Tibia DM, % | 44.38 [a] | 37.14 [c] | 42.76 [b] | 0.44 | <0.001 | 43.34 [a] | 41.56 [b] | 43.86 [a] | 0.2 | <0.001 |
| Tibia ash content, % DM | 49.86 [a] | 43.48 [b] | 48.46 [a] | 0.47 | <0.001 | 49.53 [a] | 47.02 [b] | 49.34 [a] | 0.21 | <0.001 |
| pcd of P at 21 d of age, % | 56.12 [b] | 57.15 [b] | 70.22 [a] | 0.79 | <0.001 | 47.92 [c] | 51.67 [b] | 60.36 [a] | 0.86 | <0.001 |

[a,b,c] Within a row, means without a common superscript letter differ ($p < 0.05$). PC: positive control diet; NC: negative control; PHY: NC + 500 FTU phytase/kg diet; SEM: Standard error mean; BW: body weight, ADG: average daily gain; ADFI: average daily feed intake; FCR: feed conversion ratio; cFCR: cFCR: BW gain-corrected-feed conversion ratio; EPEF: European Production Efficiency Factor; DM: dry matter; pcd of P: precaecal digestibility of phosphorus; [1] Phytase was supplemented at 500 FTU/kg diet.

Experiment 4

As shown in Table 7, birds in NC group decreased ($p < 0.001$) final BW and ADG by 2.4 and 1.6%, respectively, in comparison with birds in PC group. No significant difference was observed between PC and NC diets on ADFI, FCR and EPEF. Tibia dry matter ($-1.8\%$ points, $p < 0.001$) and tibia ash content ($-2.5\%$ points, $p < 0.001$) were lower in birds fed the NC diet compared to those fed the PC diet. In addition, birds fed NC diet exhibited higher ($p = 0.01$) pcd of P than those fed PC diet. Phytase supplementation improved the final BW, ADG, FCR and EPEF by 4.6, 4.7, 4.1 and 8.1%, respectively. In addition, tibia dry matter, tibia ash content and pcd of P were improved ($p < 0.001$) by 4.9, 2.3 and 8.7% points, respectively, in broilers fed the diet supplemented with phytase than birds on NC diet.

Combined experiments

The results on performance, bone mineralization and pcd of P and the effects of mineral reduction and phytase supplementation of the combined data obtained from all experiments are presented in Tables 8 and 9. The overall mortality was 2.8% and was not associated with dietary treatment. Compared with birds fed the adequate-nutrient diet, similarly reduction in avP and Ca by 0.15% points decreased ($p < 0.05$) the final BW, ADG, ADFI and EPEF by 6.4 ($-2.4$ to $-10.6\%$), 6.3 ($-1.6$ to $-10.7\%$), 5.9 ($-0.9$ to $-10.7\%$) and 7.1% (0.4 to $-12.8\%$), respectively, and increased ($p = 0.02$) the cFCR by 2.0% (0.3 to 3.3%). It also reduced ($p < 0.001$) the tibia dry matter and tibia ash content by 3.8 ($-1.8$ to $-7.2\%$ points) and 4.0% points ($-2.5$ to $-6.4\%$ points), respectively.

**Table 8.** Effect of phytase supplementation on growth performance, tibia traits and precaecal digestibility of phosphorus (combined trials).

| | Treatment | | | SEM | *p*-Value |
|---|---|---|---|---|---|
| | **PC** | **NC** | **PHY** [1] | | |
| Growth performance | | | | | |
| BW d1, g | 42 | 42 | 42 | 0.17 | 0.87 |
| BW d21, g | 971 [a] | 911 [b] | 990 [a] | 4.93 | 0.04 |
| ADG d1–21, g/bird | 44.12 [b] | 41.27 [c] | 45.06 [a] | 0.24 | <0.001 |
| ADFI d1–21, g/bird | 55.31 [a] | 52.86 [b] | 55.95 [a] | 0.37 | <0.001 |
| FCR d1–21 | 1.256 [ab] | 1.281 [a] | 1.248 [b] | 0.008 | 0.04 |
| Mortality 1–21 d, % | 1.68 | 1.39 | 1.48 | 0.29 | 0.78 |
| ADG d22–35, g/bird | 105.18 [a] | 98.58 [b] | 106.05 [a] | 0.56 | <0.001 |
| ADFI d22–35, g/bird | 159.37 [a] | 148.41 [b] | 160.48 [a] | 0.83 | <0.001 |
| FCR d22–35 | 1.518 | 1.509 | 1.516 | 0.007 | 0.68 |
| Mortality 22–35 d, % | 1.50 [ab] | 2.24 a | 0.93 [b] | 0.32 | 0.02 |
| BW d35, g | 2449 [a] | 2293 [b] | 2476 [a] | 21.84 | 0.04 |
| ADG d1–35, g/bird | 66.65 [a] | 62.45 [b] | 67.60 [a] | 0.35 | <0.001 |
| ADFI d1–35, g/bird | 93.54 [a] | 88.06 [b] | 94.53 [a] | 0.51 | 0.02 |
| FCR d1–35 | 1.406 | 1.412 | 1.402 | 0.005 | 0.49 |
| cFCR d1–35 [1] | 1.401 [b] | 1.429 [a] | 1.389 [b] | 0.007 | 0.001 |
| Mortality 1–35 d, % | 3.07 | 3.40 | 2.36 | 0.40 | 0.18 |
| EPEF | 460 [a] | 427 [b] | 472 a | 3.99 | <0.001 |
| *Tibia traits at 21 d of age* | | | | | |
| Tibia DM, % | 41.19 [a] | 37.42 [b] | 40.94 [a] | 0.16 | <0.001 |
| Tibia ash content, % DM | 44.20 [a] | 40.23 [b] | 44.48 [a] | 0.20 | <0.001 |
| *pcd of P at 21 d of age, %* | 57.27 [b] | 57.53 [b] | 68.62 [a] | 0.52 | <0.001 |

[a,b,c] Within a row, means without a common superscript letter differ ($p < 0.05$). PC: positive control diet; NC: negative control; PHY: NC + 500 FTU phytase/kg diet; SEM: Standard error mean; BW: body weight, ADG: average daily gain; ADFI: average daily feed intake; FCR: feed conversion ratio; cFCR: BW gain-corrected-FCR; EPEF: European Production Efficiency Factor; DM: dry matter; pcd of P: precaecal digestibility of phosphorus; [1] Phytase was supplemented at 500 FTU/kg diet.

**Table 9.** Effects of phytase on the improvements of growth performance, tibia traits and precaecal digestibility of P of broilers from 1 to 35 days of age (combined trials).

| | Comparison Relative to Control, % or % Points [1] | | | | | |
| | NC vs. PC | | | PHY vs. NC | | |
| | Means | Minimum | Maximum | Means | Minimum | Maximum |
|---|---|---|---|---|---|---|
| BW d35, g | −6.4 | −2.4 | −10.6 | +8.0 | +4.6 | +11.4 |
| ADG 1–35 d, g/bird/d | −6.3 | −1.6 | −10.7 | +8.3 | +4.7 | +10.7 |
| ADFI 1–35 d, g/bird/d | −5.9 | −0.9 | −10.7 | +7.3 | +0.5 | +12.3 |
| cFCR 1–35 d | +2.0 | +0.3 | +3.3 | −2.8 | +1.2 | −6.5 |
| EPEF | −7.1 | +0.4 | −12.6 | +10.6 | +8.1 | +14.5 |
| Tibia DM, % [1] | −3.8 | −1.8 | −7.2 | +3.5 | +2.1 | +4.7 |
| Tibia ash content, % DM [1] | −4.0 | −2.5 | −6.4 | +4.2 | +2.3 | +5.7 |
| pcd of P at 21 d of age, % [1] | +0.3 | −2.1 | +3.7 | +11.1 | +6.3 | +18.6 |

PC: positive control diet; NC: negative control; PHY: NC + 500 FTU phytase/kg diet; SEM: Standard error mean; BW: body weight, ADG: average daily gain; ADFI: average daily feed intake; cFCR: BW gain-corrected-feed conversion ratio; EPEF: European Production Efficiency Factor; pcd of P: precaecal digestibility of phosphorus; [1] The improvement was expressed as % points.

No significant difference was observed in pcd of P between the PC and NC diets. Birds fed PHY diets improved ($p < 0.05$) the final BW, ADG, ADFI, EPEF and cFCR by 8.0 (4.6 to 11.4%), 8.3 (4.7 to 10.7%), 7.3 (0.5 to 12.3%), 10.6 (8.1 to 14.5%) and 2.8% (1.2 to −6.5%), respectively. Phytase addition at 500 FTU/kg diet also increased ($p < 0.001$) the tibia dry matter and tibia ash content by 3.5 (2.1 to 4.7% points) and 4.2% points (2.3 to 5.7% points), respectively. In addition, phytase supplementation improved the pcd of P ($p < 0.001$) by 11.1 and 11.3% points, in comparison with NC and PC diets, respectively. All performance parameters, tibia dry matter, and tibia ash content obtained in NC diet supplemented with phytase were comparable to or better than the positive control diet.

## 4. Discussion

Phytase enzyme is commonly used as a feed additive in diets to release phytate-bound P, thereby improving growth performance and bone mineralization in broilers and reducing the need for mineral phosphate to meet the P requirement of birds. However, the efficacy and consistency of such phytase on these former parameters can vary depending on several factors such as the intrinsic characteristics of phytase (source, dose, optima pH and its affinity to phytate) and diet (composition and the content of phytate, Ca and avP). As expected, the NC treatments with avP- and Ca-deficient diets resulted in lower feed intake and, consequently, lower ADG and final BW of broilers compared to the PC diet formulated to meet or exceed the nutrient requirements for birds. These results agree with Sharma et al. [20] and Zhang et al. [21], who reported adverse effects of dietary avP and Ca deficiency on growth performance in broilers. Indeed, P is the second most abundant mineral after calcium in body composition. Therefore, it is an essential nutrient required for normal growth [22,23], and it plays vital roles in maintaining osmotic and acid-base balance, energy metabolism, amino acid metabolism and protein synthesis [24,25]. Otherwise, several studies performed on broilers have reported improvements in performance when supplemental phytase was used [2,5,11,13,21,24,26]. In the present study, the supplementation of the NC diets with the novel bacterial 6-phytase at 500 FTU/kg diet improved the final BW, ADG, ADFI, EPEF and cFCR, confirming at least the availability of P and Ca due to phytase addition to diet reduced in P and Ca. Similarly, Walters et al. [27] reported beneficial effects of phytase addition at 500 FTU/kg diet on growth performance of broilers fed P- and Ca-reduced diets. Furthermore, according to the results obtained on performance, a consistently positive effect of phytase supplementation was observed among all trials. These results strongly suggest that this novel phytase has the ability to counteract the adverse effects of P and Ca depletion and be efficient under different experimental con-

ditions, including variability among feed ingredients. They also highlight the consistent positive effect of this new bacterial phytase (Table 9) in broiler production.

In addition to performance, nutritional P and Ca deficiency can impair bone integrity [28]. Several studies reported that bone ash is considered the most sensitive indicator of P and Ca status and a useful response criterion to estimate the P equivalency of exogenous phytases [29–31]. In the current study, when birds were fed the diets reduced similarly in avP and Ca by 0.15% points below the nutrient recommendations for broilers, there was a reduction in tibia dry matter and tibia ash content. These results are in accordance with the findings of Onyango et al. [32], who demonstrated that bone ash content was linearly decreased as the dietary level of Ca decreased. However, phytase addition at 500 FTU/kg diet was required to restore former tibia traits to the levels of the PC diet, thereby confirming an increase in the availability of P and Ca in the NC diet supplemented with phytase. This finding agrees with Dersjant-Li et al. [33], who reported a significant improvement in tibia ash content with phytase at 500 FTU/kg diet in broilers fed corn–soybean meal-based diet reduced in nutrients. The absence of response observed on pcd P in birds fed diets reduced in minerals compared with bird fed the adequate-nutrient diet has also been reported previously. Indeed, Yan et al. [34] and Rousseau et al. [35] reported that birds could adapt to the Ca and P deficiencies by increasing the degradation of phytate-P in the small intestine through upregulation of the main intestinal P transporters or by increasing the endogenous phosphatases activities [36,37]. Hence, the birds have the ability to compensate for the dietary mineral reduction by increasing their P digestibility. Rodehutscord et al. [38] reported a substantial ability of broilers to degrade phytate (68% in average) and digest P even without any exogenous phytase supplementation probably by the action of endogenous (mucosal and microbial) phytases presented in the gut intestinal tract. As reported by Tamim er al. [39], dietary Ca level can reduce the inherent ability of broilers to degrade phytate and absorb P. However, the relationship between Ca level and the capability of endogenous phytases to degrade phytate remains not fully elucidated.

In the present study, the phytase supplementation at the standard dose improved the pcd of P by approximately 11.1% points, probably due to the degradation of phytate by phytase. Many studies have observed improvements in P availability in broilers fed low-P diets supplemented with phytase at 500 FTU/kg diet [5,13,33,40]. It has been reported that early and thorough hydrolysis of phytate by exogenous phytase in the upper digestive tract is crucial for enhancing the release of P, and perhaps of energy and amino acids, thus limiting the anti-nutrient effects of phytate within the intestinal tract [7,11]. It is also worth to indicate that the result observed on pcd of P with phytase in the present study was clear proof of improvements observed on growth performance and bone mineralization. However, further studies are needed to evaluate the effect of the present phytase at different doses on pcd P and growth performance depending on dietary levels of Ca and phytate.

## 5. Conclusions

In conclusion, this study showed that the novel bacterial 6-phytase added at 500 FTU/kg diet allowed birds fed diets reduced in P and Ca to reach a growth performance equivalent to that of birds fed a diet adequate in all nutrients in a more economically and sustainable way. Thus, available P and Ca can be lowered similarly by 0.15% points in phytase-supplemented diets without compromising feed efficiency and reducing the production cost of broilers. Further studies need to be carried out on broilers to clarify the side effects of this new bacterial phytase beyond performance, bone mineralization and nutrient digestibility.

**Author Contributions:** Conceptualization, M.J., C.H. and A.P.; methodology, M.J., A.P. and P.C.; software, M.J. and P.C.; validation, M.J., C.H., M.C., B.Y., S.O., A.P. and E.D.; formal analysis, M.J.; investigation, M.J., P.C. and A.P.; writing-original draft preparation, M.J.; writing-review and editing, M.J., C.H., M.I.G., F.K., M.F., T.R., G.C.L., P.C., M.C., B.Y., S.O., A.P. and E.D. All authors have read and agreed to the published version of the manuscript.

**Funding:** This research received no external funding.

**Institutional Review Board Statement:** The experimental procedures were approved by the Animal Care and Use committees at IRTA Animal Nutrition (number 11421, Spain), IMASDE Agroalimentaria (RD 1084/2005, Spain), Center of Expertise in Research and Nutrition (G-03-159-4, France) and Scotland's Rural College (POUAE 08-2021, UK).

**Data Availability Statement:** Data can be available requiring via email to Maamer Jlali, maamer.jlali@adisseo.com.

**Conflicts of Interest:** The authors declare there are no known conflict of interest associated with this publication and there has been no significant financial support for this work that could have influenced its outcome.

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
