# Peer review of "A Novel Bacterial 6-Phytase Improves Growth Performance, Tibia Mineralization and Precaecal Digestibility of Phosphorus in Broilers: Data from Four Independent Performance Trials"

_agriculture, doi:10.3390/agriculture13081507_

Round 1
Reviewer 1 Report
The authors have written the manuscript quite well; however, I still have the following questions:
1) What is the rationale behind conducting four experiments with a common objective in this study?
2) The authors should provide further rationale for their interest in this enzyme and why did the authors choose to supplement the enzyme at a level of 500 FTU/kg? How did the authors determine that this level was appropriate for enzyme supplementation in the diet?
3) Why did the authors select male broilers for this experiment? Can the findings of this experiment be extrapolated for use in female broilers?
Reviewer 1 Report
The authors have written the manuscript quite well; however, I still have the following questions:
1) What is the rationale behind conducting four experiments with a common objective in this study?
2) The authors should provide further rationale for their interest in this enzyme and why did the authors choose to supplement the enzyme at a level of 500 FTU/kg? How did the authors determine that this level was appropriate for enzyme supplementation in the diet?
3) Why did the authors select male broilers for this experiment? Can the findings of this experiment be extrapolated for use in female broilers?
Author Response
Thank you for your comments.
1) What is the rationale behind conducting four experiments with a common objective in this study?
Au: The rationale behind conducting four experiments with a common objective was to evaluate the consistency and the reliability of the efficacy of the new phytase to improve growth performance, bone mineralization and precaecal digestibility of P in different facilities.
2) The authors should provide further rationale for their interest in this enzyme and why did the authors choose to supplement the enzyme at a level of 500 FTU/kg? How did the authors determine that this level was appropriate for enzyme supplementation in the diet?
Au: 500 FTU/kg diet is usually used as standard dose for such exogenous phytase. We did some previous studies to determine the P equivalent of our phytase at different doses and we observed that our phytase at 500 FTU/kg diet was able to replace 0.15% point of available phosphorus (avP). This was the reason why we used 500 FTU/kg diet in the present study.
3) Why did the authors select male broilers for this experiment? Can the findings of this experiment be extrapolated for use in female broilers?
Au: Usually we used male broilers in order to have less variability between birds and to have a good uniformity within a flock. In my opinion, the findings of this experiment can be extrapolated to female broilers. Sure, we expect a difference between male and female chickens in terms of performance (final BW, ADG and FCR) but not in terms of efficacy of phytase and precaecal digestibility of phosphorus. Sebastian et al. (1996; Poultry Science 75:729-736) did not observe a significant interaction between diet (included those supplemented with phytase) and sex. Mirabile et al. (2020; Animal Feed Science and Technology 270: 114649) reported that the sex of broilers does not influence the true ileal P digestibility coefficient in soybean meal, one of the main feed ingredients in broiler diets.
Reviewer 2 Report
Line 2: I would highly recommend not to use a dose in a title
Line 96: Why were MCP was used in experiments 2-4 and DCP in experiment 1? Could the effects obtained be due to the different bioavailability of P from MCP and DCP?
Line 186: Please include the total numer of birds sacrified in each experiment
Line 191: The same, clarify how many tibias were collected and intended for further analysis? Was the numer the same in each experiment?
Line 338: It needs to be clarified whether consent was obtained from the ethics committee for the country concerned in each experiment (if so, the consent number), or whether consent was waived in the experiments by considering the rearing of broiler chickens as an farming activity not requiring procedures to be performed on live animals.
The discussion should be expanded to include elements of novelty in the phytase used. Authtors state that it is a novel product. How then does it differ from other phytases? Is the effect of using this phytase better than others? What indicators support this? The effects of the phytase are known, it would be necessary to compare the authors' results with other publications on phytases (e.g. phytases from other bacterial origins).
The paper should have specified why 4 experiments with a similar experimental set-up were performed. Is it about the applicability of the product in different geographical locations? Is it the versatility for use in different countries where the composition of feed components may differ?
Reviewer 2 Report
Line 2: I would highly recommend not to use a dose in a title
Line 96: Why were MCP was used in experiments 2-4 and DCP in experiment 1? Could the effects obtained be due to the different bioavailability of P from MCP and DCP?
Line 186: Please include the total numer of birds sacrified in each experiment
Line 191: The same, clarify how many tibias were collected and intended for further analysis? Was the numer the same in each experiment?
Line 338: It needs to be clarified whether consent was obtained from the ethics committee for the country concerned in each experiment (if so, the consent number), or whether consent was waived in the experiments by considering the rearing of broiler chickens as an farming activity not requiring procedures to be performed on live animals.
The discussion should be expanded to include elements of novelty in the phytase used. Authtors state that it is a novel product. How then does it differ from other phytases? Is the effect of using this phytase better than others? What indicators support this? The effects of the phytase are known, it would be necessary to compare the authors' results with other publications on phytases (e.g. phytases from other bacterial origins).
The paper should have specified why 4 experiments with a similar experimental set-up were performed. Is it about the applicability of the product in different geographical locations? Is it the versatility for use in different countries where the composition of feed components may differ?
Author Response
Thank you for your comments
Line 2: I would highly recommend not to use a dose in a title
Au: The dose of phytase was removed from the title
Line 96: Why were MCP was used in experiments 2-4 and DCP in experiment 1? Could the effects obtained be due to the different bioavailability of P from MCP and DCP?
Au: Depending on the availability of inorganic P source in the facility, MCP or DCP was used. However, the difference between MCP and DCP in terms of bioavailability of P was considered during feed formulation.
Line 186: Please include the total numer of birds sacrified in each experiment
Au: Done in lines 192 to 194.
Line 191: The same, clarify how many tibias were collected and intended for further analysis? Was the numer the same in each experiment?
Au: Done in lines 198 to 200
Line 338: It needs to be clarified whether consent was obtained from the ethics committee for the country concerned in each experiment (if so, the consent number), or whether consent was waived in the experiments by considering the rearing of broiler chickens as an farming activity not requiring procedures to be performed on live animals.
Au: Done in lines 421 to 423.
The discussion should be expanded to include elements of novelty in the phytase used. Authtors state that it is a novel product. How then does it differ from other phytases? Is the effect of using this phytase better than others? What indicators support this? The effects of the phytase are known, it would be necessary to compare the authors' results with other publications on phytases (e.g. phytases from other bacterial origins).
Au: Done in discussion part
The paper should have specified why 4 experiments with a similar experimental set-up were performed. Is it about the applicability of the product in different geographical locations? Is it the versatility for use in different countries where the composition of feed components may differ?
Au: Done at the end of the introduction and also in the discussion part.
Reviewer 3 Report
The topic is very interesting for poultry breeders and researchers, the objectives of the study are clear. However, the introduction is too short. It should contain more details of the scientific background of the use of phytase. The experimental design is correct, but it would be advisable to see the statistical model with the variables. I recommend to demonstrate the BW, ADG and FCR with graphs, and maybe extend the discussion in more details.
-
Reviewer 3 Report
The topic is very interesting for poultry breeders and researchers, the objectives of the study are clear. However, the introduction is too short. It should contain more details of the scientific background of the use of phytase. The experimental design is correct, but it would be advisable to see the statistical model with the variables. I recommend to demonstrate the BW, ADG and FCR with graphs, and maybe extend the discussion in more details.
Author Response
Thank you for your comments.
The topic is very interesting for poultry breeders and researchers, the objectives of the study are clear. However, the introduction is too short. It should contain more details of the scientific background of the use of phytase.
Au: More details of the scientific background of the use of phytase were added in the introduction (lines 62 to 67).
The experimental design is correct, but it would be advisable to see the statistical model with the variables.
Au: The statistical model with the variables was added in lines 227 to 230.
I recommend to demonstrate the BW, ADG and FCR with graphs, and maybe extend the discussion in more details.
Au: The discussion was extended. However, performance parameters (BW, ADG and FCR) were kept as tables.